# Pulmonary Langerhans Cell Histiocytosis in an African Lion: A Rare Case Report

**DOI:** 10.3390/ani14071011

**Published:** 2024-03-26

**Authors:** Liang Zhang, Hui Chen, Yulin Ding, Wenlong Wang, Gao Wa, Bingwu Zheng, Jinling Wang

**Affiliations:** 1College of Veterinary Medicine, Inner Mongolia Agricultural University, Hohhot 010018, China; zhangliang2018@imau.edu.cn (L.Z.); chenhui2018@imau.edu.cn (H.C.); dingyulin2001@126.com (Y.D.); wwl.imau@163.com (W.W.); 2Key Laboratory of Clinical Diagnosis and Treatment Techniques for Animal Disease, Ministry of Agriculture and Rural Affairs, Hohhot 010018, China; 3Inner Mongolia Autonomous Region Key Laboratory of Veterinary Fundamentals and Disease Prevention and Control of Herbivorous Livestock, Hohhot 010018, China; 4Inner Mongolia Autonomous Region Key Laboratory of Tick-Borne Zoonotic Infectious Disease, Bayan Nur 015000, China; melody_gaowa@163.com; 5Department of Medicine, College of Hetao, Bayan Nur 015000, China; 6Hohhot Zoo, Hohhot 010050, China; zhengbw119@126.com

**Keywords:** African lion, lung, Langerhans cell histiocytosis, immunohistochemistry

## Abstract

**Simple Summary:**

Feline pulmonary Langerhans cells histiocytosis (PLCH) is a rare disease that results in respiratory failure due to the infiltration of Langerhans cells (LCs) in the lungs. A diagnosis of PLCH is proposed based on the clinical signs and pathological findings and confirmed based on the infiltrating histiocytic cells. Here, we present a rare case of PLCH in an African lion. The disease mainly affected the lungs with variable and limited involvement of other organs. While a definite clinical diagnosis is difficult, the pulmonary pathological changes with nodular appearance and infiltration of LCs, together with the immunohistochemistry results, are sufficient for diagnostic confirmation of PLCH. To our knowledge, this is the first reported PLCH case in an African lion, and this case report could present new information and aspects of this feline histiocytic disease.

**Abstract:**

Background: Feline pulmonary Langerhans cells histiocytosis (PLCH) is a rare disorder that results in progressive respiratory failure secondary to pulmonary parenchymal infiltration with Langerhans cells (LCs). A diagnosis of PLCH is proposed based on the clinical features and pathological findings and confirmed based on the infiltrating histiocytic cells. There are few documented cases of feline PLCH, and this case report of PLCH in an African Lion could present new information and aspects of this feline histiocytic disease. Case presentation: An African lion at Hohhot Zoo showing severe hyporexia and dyspnea with subsequent mental depression and emaciation died of exhaustion after a 35-day course of illness. Empirical treatment did not have a significant effect. An autopsy revealed that the lungs were enlarged and hardened due to infiltrative lesions, with many yellowish-white foci in all the lobes and sections. Furthermore, the kidneys were atrophied and had scattered grayish-white lesions on the surface. At the same time, congestion was widely distributed in various locations, including the liver, subcutaneous loose connective tissues, serosal surface and other tissues and organs. Histologically, proliferative histiocytic cells (PHCs) were scattered in the alveolar cavities, bronchioles and submucosa of bronchioles, with evident cellular and nuclear pleomorphism, and thus the alveolar septa were obliterated. The histopathological changes in other organs included chronic sclerosing glomerulonephritis, proliferated Kupffer cells in the liver, adrenal edema and interstitial connective tissue hyperplasia, as well as atrophy of the small intestines and spleen. Furthermore, immunohistochemical analysis results were strongly positive for CD1a, vimentin, S100 and E-cadherin in the membrane or cytoplasm of PHCs, supporting an LC phenotype. Conclusions: Here, we present a rare pulmonary Langerhans cell histiocytosis case in an African lion.

## 1. Introduction

Langerhans cells (LCs) are antigen-presenting dendritic cells derived from bone marrow that colonize the epidermis of the skin and other epithelia of multiple mucosal sites (pulmonary bronchi, oropharynx, tongue, ectocervix and vagina) [1]. In the lungs, LCs are located in the bronchial epithelium. Disorders related to LC proliferation are called LC histiocytosis (LCH) and have been documented in human and veterinary medicine, and are mainly best described in humans [2,3]. Human LCH occurs primarily in children and covers a range of diseases that can be divided into single-system and multisystem LCH [2,4]. Single-system LCH is characterized by single or multiple lesions in one organ, such as the lymph nodes, skin or bone. At the same time, multisystem LCH causes lesions in two or more organs, including the lymph nodes, pituitary gland, lungs, spleen, liver, skin, bone and bone marrow. However, in veterinary medicine, proliferative disorders of LC have not been extensively documented. The best-described entity is cutaneous histiocytoma (a tumor of epidermal LC) in canine animals such as dogs. In contrast, histiocytic proliferative diseases of an LC origin in felines are generally uncommon and consist mainly of histiocytic sarcoma, feline progressive histiocytosis and pulmonary LCH (PLCH) [3,5].

Taking cats as an example of felines, PLCH is a rare disorder in aged animal individuals (10–15 years) that results in progressive respiratory failure secondary to pulmonary parenchymal infiltration with LCs, generally leading to death or euthanasia [6,7]. Like human PLCH, feline PLCH has been characterized as a pulmonary disease alone or with multisystem involvement, including the lungs, thyroid, parathyroid gland, liver, pancreas, spleen, kidneys and mesenteric, hepatosplenic and tracheobronchial lymph nodes [3,6,8]. The diagnosis of PLCH is suspected mainly based on the clinical features and pathological findings and subsequently confirmed on the basis of the infiltrating histiocytic cells’ immunohistochemical (IHC) profile. Clinically, cats with this disease often have severe respiratory distress characterized by tachypnea, apparent respiratory effort or open-mouth breathing [3,5]. Gross findings are marked by multinodular-to-diffuse involvement in all pulmonary lobes and occasionally in other organs [3,8]. Histologically, lesions are manifested as cohesive infiltrating histiocytic cells, with evident cellular and nuclear pleomorphism, which obliterate the terminal airways, peri-bronchial parenchyma and alveolar septa and possibly extend to pleural surfaces and extrapulmonary sites [3,6]. It should be noted that gross and histological changes are frequently inadequate to confirm the diagnosis of PLCH, and IHC is an effective method to identify the originating cells in such cases. Specifically, LCs could be determined by the expression of molecules such as CD1a, CD18, vimentin, E-cadherin, etc. [3,7,9].

In the present study, we describe the clinical, gross, histological and IHC features of a novel rare case of histiocytic proliferative disease of LC origin in an African Lion, which primarily affected the lungs, with limited involvement of other organs. To our knowledge, this is the first report of a PLCH case in an African Lion, a member of the feline family, presenting new aspects of a poorly documented feline histiocytic disease.

## 2. Case Presentation

### 2.1. Clinical Features

A 10-year-old male African lion showed hyporexia, gradual mental depression and slow movements at Hohhot Zoo in Hohhot, Inner Mongolia Autonomous Region, on 25 August 2022. After daily empirical treatment of intramuscular injections of vancomycin, cefoperazone sodium and dexamethasone with an anesthetic gun, as well as oral administration of vitamins with minerals tablets, the patient expressed occasional appetite, but still with little meat consumption. With development of the disease, the lion’s back was arched, and it was gradually reluctant to walk, preferring to lie down, with progressive wasting, breathing difficulties and abdominal breathing. By 26 September 2022, the animal was lying on the ground, unable to rise, head drooping and severely emaciated, and had severe dyspnea with shallow and fast breathing. Due to the progressive worsening of symptoms, daily intramuscular injections of compound sulfadiazine sodium, ATP and coenzyme A were given to the sick lion in the late stage of the disease, but it was ineffective. On 27 September, after anesthesia with xylazine hydrochloride injection, the lion showed mild cyanosis of visual mucosa, dyspnea and severe emaciation, weighing only 104.5 kg (normal body weight being about 200 kg). Dexamethasone, ATP, Coenzyme A, penicillin, vitamin C, creatinine, sodium bicarbonate, 0.9% saline, 5% glucose solution and epinephrine were injected intravenously, still without any therapeutic effect. Finally, he died of exhaustion on 28 September 2022, with the course of disease lasting for 35 days.

### 2.2. Gross Pathology

The dead African lion was submitted to a complete necropsy. The corpse was severely emaciated, and extensive alopecic areas were present (Figure 1). There was a lot of stale yellow and translucent liquid accumulated in the chest cavity. The lungs were enlarged, with consolidation in all the diaphragmatic and accessory lobes and most of the apical and cardiac lobes. The areas of consolidation were grayish-white or grayish-red in color, presenting a shiny flesh-like transformation (Figure 2A). Sections of the lung were also grayish-white or grayish-red and relatively rough, with visible stale yellowish-white foci of varying sizes (Figure 2B). There was a white foamy liquid in the trachea (Figure 2C). Severe eccentric hypertrophy of the heart was seen, with noticeable widening of the transverse diameter of the heart, blunting of the apex, and significant ventricular dilation (Figure 3A). In particular, severe dilation of the right ventricle and thickening of the heart wall were found.

In addition, the systemic subcutaneous loose connective tissue and serosal surface exhibited congestion, and the veins presented dilation. Complete atrophy and disappearance of subcutaneous and abdominal fat were seen. The liver was dark red in color, with a soft texture and a large amount of dark red blood flowing out from the section (Figure 3B).

The spleen presented with a soft texture and a dark purplish red surface (Figure 3C). The volumes of the bilateral kidneys were decreased, with scattered slightly concave grayish-white lesions on the surface with varying sizes and shapes (Figure 3D). The gastric volume was severely reduced without contents, and the serosal veins were remarkably dilated, resulting in severe congestion. Moreover, other tissues and organs showed congestion.

Taken together, the gross lesions showed that the main manifestations of the dissected animal were lung enlargement and consolidation in most areas of the lungs, with heart failure and atrophy of other organs.

### 2.3. Histopathology

The main histopathological features of the lung included scattered or dense proliferative histiocytic cells (PHCs) in the bronchioles at all levels and in the alveolar cavities, as well as connective tissue hyperplasia in the interstitium (Figure 4A,B). The lobular, bronchiolar and alveolar structures of the lung disappeared and were filled with many PHCs (Figure 4A). The pulmonary septal spaces in the consolidation areas were markedly widened, caused by severe fibrosis in the interstitium (Figure 4B). PHCs had a large volume and were circular, elliptical or irregular in shape (Figure 4C). Most of such cells exhibited pseudopodial protrusions from their surfaces, connecting them. The cytoplasm was rich and pale pink, with some cells showing vacuolar lipid droplets in the cytoplasm. The nuclei were circular, elliptical or reniform, with clearly visible nucleoli, and a mitotic appearance of the nuclei was occasionally seen, as well as binucleated cells (Figure 4C,D). Type II alveolar epithelial cells showed mild or distinct proliferation, with partial detachment of bronchial epithelial cells at all levels, and scattered infiltration of neutrophils and lymphocytes was present in the alveolar cavity (Figure 4D). There was edema in the submucosa of bronchioles, where PHCs were scattered. Some areas of lung tissue displayed diffuse necrosis, forming interconnected necrotic foci with varying sizes and shapes.

Furthermore, mild congestion and dilation of the small blood vessels in the myocardial interstitium were seen, with a demonstrable deposition of brownish yellow lipofuscin particles in the cytoplasm of myocardial cells (Figure 5A). Kupffer cells in the liver were widely activated and proliferated, engulfing many hemosiderin-containing particles. Severe congestion and a hemorrhage around the central vein were observed in the middle lobe of the liver (Figure 5B). The number of lymphocytes in the spleen was significantly reduced, and the volume of white pulp was small. Iron-swallowing cells were widely distributed in the red pulp, with apoptotic lymphocytes visible (Figure 5C). The histopathological changes in the kidneys were characterized by chronic sclerosing glomerulonephritis, diffuse fibrosis in the renal cortex, evident hyperplasia of the glomerular capsule and the surrounding connective tissue and complete fibrosis of the entire glomerulus in severe areas. The residual renal tubular epithelial cells were swollen and exhibited slight to moderate steatosis, and scattered or more abundant lipofuscin particles could be seen in the cytoplasm (Figure 5D). The walls and villi of the small intestine were thinner than usual. Other findings included edema and interstitial connective tissue hyperplasia of the adrenal gland.

According to the diffuse proliferation and the fibrosis and extensive necrosis of cells with histiocytic characteristics restricted in lung tissue, together with heart failure and congestion and atrophy of other tissues, the illness can be preliminarily diagnosed as lung histiocytosis.

### 2.4. Immunohistochemistry

IHC analysis with diaminobenzidine staining and with all monoclonal antibodies (Abcarta) showed positive reactions for the monoclonal antibodies to CD1a (PA538; Figure 6A) in the membrane, and Vimentin (PA040; Figure 6B), S100 (PA139; Figure 6C) and E-cadherin (PA073; Figure 6D) in the cytoplasm of PHCs. Adverse reactions were observed for the monoclonal antibody to CD68. These data indicate that the PHCs were derived from LCs.

## 3. Discussion

Rare cases of histiocytosis have been reported in cats, but to our knowledge, there are no known cases in African lions. In the current case of the African lion, the diagnosis of PLCH was made according to the pathological findings and the IHC profile of the proliferating cells. In cats, the occurrence of PLCH is associated with the age of animals, usually found in middle-aged to older ones. Although the sample size is insufficient to provide epidemiological evidence, the patient mentioned in the present study was a 10-year-old African lion (lifespan of 10–15 years), similar to what other researchers reported in cats [8,10].

Several reported PLCH cases in cats presented similar clinical signs, primarily with severe respiratory complaints, anorexia, lethargy and weight loss, with particularly severe breathing difficulties [3,8,10]. In the case presented here, the patient showed dyspnea and abdominal breathing, along with mental depression, loss of appetite, severe emaciation and even the inability to stand; these symptoms are consistent with feline PLCH cases. Respiratory symptoms are caused by the widespread involvement of the pulmonary parenchyma, resulting in progressive respiratory disease characterized by mixed restrictive dyspnea that eventually leads to death [8,10]. There is little information associated with the treatment of feline PLCH. In some cases, supportive therapy was initiated, but no clinical response was seen, and ultimately all cats died [8]. Bronchodilators, corticosteroids and diuretics were previously commonly administered in other cases, but these treatments also tended to be unsuccessful [10]. Our empirical therapy with vancomycin, cefoperazone sodium, dexamethasone, vitamins with minerals tablets or compound sulfamethoxazole sodium, ATP and coenzyme A, did not have a significant therapeutic effect. Moreover, it has been reported that the symptoms of PLCH in cats can be acute or present for several months [3]. The clinical course in the current case was 35 days, reflecting a chronic clinical progression.

Gross findings in cases of PLCH are marked by infiltrative nodules affecting all lung lobes [6,8,11]. Initially, multifocal nodules correspond histologically to the involvement of the peri-bronchial parenchyma but tend to merge and subsequently affect the pleural surface [3,6,8]. Similarly, infiltrative lesions in the lungs were observed in the African lion. Specifically, the lungs were enlarged and hardened with a shiny, flesh-like transformation, with many clear yellowish-white foci observed in all the lobes and sections. Furthermore, a quantity of abnormal or pathological liquid had accumulated in the chest cavity and the trachea, pointing to underlying lesions in the pleural surface and tracheal or bronchial tissues. In addition, it was previously reported that minor involvement of the lymph nodes, thyroid, parathyroid gland, kidney, liver and spleen can be observed in cat PLCH cases [8]. In the present case of the African lion, the kidneys were atrophied and showed scattered grayish-white lesions on the surface. At the same time, there was widely distributed congestion in various locations, including the liver, subcutaneous loose connective tissues, serosal surface and other tissues and organs. These pathological changes were consistent with those described in the relatively few cases of internal LCH with hepatosplenomegaly, ascites and extensive metastasis affecting the esophagus, kidneys and pancreas [8,12].

Severe eccentric hypertrophy of the heart was observed, with significant ventricular dilation, especially in the right ventricle. Such cardiac lesions have scarcely previously been reported in other animal species, and the possible reasons could be complex and may involve systemic lesions in this patient. The extensive pulmonary consolidation caused by diffuse proliferation of PHCs and proliferation and fibrosis of type II alveolar epithelial cells resulted in pulmonary hypertension, which led to the obstruction of right ventricular blood flow into the lungs. Such an obstruction resulted in an overload of the blood and pressure on the right side of the heart, gradually causing compensatory hypertrophy of the entire organ, especially of the right side. While the compensatory limit was exceeded, right heart failure occurred, that is, pulmonary heart disease, and congestion, edema and cellular damage was present in the tissues and organs throughout the whole body.

Feline PLCH led to the obliteration of terminal airways and contiguous alveoli by cohesive sheets of histiocytes with the immunophenotypic and ultrastructural features of LC [10]. Histocytologically, the infiltrating histiocytic cells had obvious cellular pleomorphism and a variable nuclear morphology, ranging from mature to immature, with the peri-bronchial parenchyma, alveolar septa and alveolar spaces being obliterated [3,8,10]. Similarly, in this case, proliferative histiocytic cells scattered in the alveolar cavities, bronchioles and submucosa of bronchioles presented as circular, elliptical or irregular in shape, and the nuclei were circular, elliptical or reniform, with a mitotic and binucleated appearance occasionally observed. In addition, the pulmonary septa were markedly widened, probably due to the infiltration of proliferating histiocytes. The obliteration of alveolar spaces by intense and characteristic histiocytic infiltrates is one of the causes of lesions in the lung parenchyma [6,8]. As previously reported, pulmonary smooth muscle hypertrophy in affected bronchioles and fibrosis in the adjacent alveolar interstitium, in addition to aggregates of lymphocytes and fewer plasma cells scattered primarily on the perivascular, subpleural and peribronchial interstitium, were also frequently observed [6]. We also found that scattered infiltration of neutrophils and lymphocytes was presented in the alveolar cavity, which is consistent with previous studies. A crucial role of T lymphocytes in the development and progression of LCH has been suggested [13]. A great variety of cytokines produced by T lymphocytes and LCs have been detected in human LCH, leading to the “cytokine storm” hypothesis, which assumes that T lymphocytes and LCs participate in a cytokine amplification cascade, thus affecting the recruitment, maturation and proliferation of LCs in LCH diseases [13,14].

In humans, multiorgan disease in adult PLCH occurs in up to 20% of cases [15,16]. Moreover, the involvement of multiple organs in feline PLCH has also been reported as histiocytic infiltration of the liver, kidneys and pancreas, and tracheobronchial or mesenteric lymph nodes in several cat cases were observed via gross or microscopic examination [6]. In this case, the histopathological changes in the kidneys were mainly highlighted as chronic sclerosing glomerulonephritis with diffuse fibrosis in the renal cortex and evident hyperplasia of the glomerular capsule, possibly caused by infiltration of proliferating histiocytes. Additionally, Kupffer cells in the liver were evidently proliferated, and edema and interstitial connective tissue hyperplasia were observed in the adrenal gland. The wall and villi of the small intestine had thinned, and the white pulp of the spleen had shrunk significantly. Scattered lipofuscin particles could be seen in the cytoplasm of the residual renal tubular epithelial cells and myocardial cells, indicating an enhanced oxidative response in these cells, leading to cellular aging and damage [17]. In felines, the presence of extrapulmonary lesions, accompanied by enhanced anisokaryosis in extrapulmonary sites such as tracheobronchial lymph nodes, is a feature that favors neoplasia transformations [10]. However, extrapulmonary lesions are observed in human PLCH, which is primarily considered a reactive disease.

Diagnostic confirmation in cases of PLCH in humans and veterinary medicine depends on the IHC profile of histiocytic cells [3,6,8]. Langerhans cells are confirmed in tissues by their expression of CD1a, CD18, vimentin, E-cadherin, Iba-1, major histocompatibility complex class II molecules and langerin (CD207) [3,6,8,18]. Following previous reports of pulmonary LCH, the IHC analysis results in this study showed that the PHCs were strongly positive for CD1a, vimentin, S100 and E-cadherin, as well as negative for CD68, which supports an LC phenotype. The CD1a molecule is expressed exclusively by LCs; it could be used to confirm further the diagnosis of feline LCH, and its positive expression in the present study is consistent with an LC origin. Although some researchers purport that the availability and use of CD1a are limited because the antibody is not assessable in formalin-fixed tissues [6], Peter Moore et al. argued that its expression has been confirmed in frozen sections of lung lesions in a recent case [3]. Vimentin is overexpressed in various epithelial cancers [19], and S100 is widely reported to participate in multiple signaling pathways in tumor cells [20]. In our study, the positive expression of the two molecules in the PHCs could be considered as evidence in favor of neoplastic transformations. The adhesion molecule E-cadherin is mainly expressed by epithelial cells, but also by LCs, and LCs are the only histiocytic cells that express E-cadherin [5,8,10]. Given that E-cadherin immunolabeling has not been reported in other histiocytic disorders of canines and felines, its positivity is highly supportive of an LC origin [16,21]. Therefore, the positive result of our IHC labeling for E-cadherin, in association with those for CD1a, vimentin and S100, further confirms the diagnosis of an LC origin of lung histiocytosis. Furthermore, the myeloid marker CD68 is a protein highly expressed in circulating and tissue macrophages [22], and it is traditionally employed to immunostain monocytes/macrophages in the histochemical analysis of inflamed tissues and tumor tissues [23]. Therefore, in this study, the negative expression of this molecule in the PHCs indicated that the infiltrated cells could be ruled out as having a monocyte or macrophage origin.

The potential pathogenesis of LCH still needs to be clearly understood in humans and animal species. Although some researchers tend to consider it as a reactive disorder, others suggest that it is a neoplastic transformation rather than a functional deregulation of LCs, due to the characteristic of LCH is tissue infiltration with LCs that exhibit varying degrees of pleomorphism [2,3,4,10]. Moreover, studies have identified many oncogenic mutations in affected human patients, supporting the latter hypothesis [15]. Comparably, PLCH represents a neoplastic process because of its cellular morphological characteristics and extrapulmonary lesions in cats [10]. LCH is currently defined as an inflammatory myeloid neoplasm in the revised 2016 Histiocyte Society classification [15]. PLCH in humans is strongly associated with tobacco smoke and, less often, other elements such as genetics and other factors, indicating that different mechanisms may also produce a marked effect in the occurrence and development of this disease [2,4,15]. In this study, there was no history of tobacco smoke or other pulmonary irritant exposure for the African lion with PLCH. Interestingly, Argenta et al. reported that a cat with PLCH had been exposed to tobacco smoke, but the significance of this finding remains elusive [8]. To provide insights into the pathogenesis of this enigmatic disorder, i.e., PLCH, the development of molecular techniques to assess clonality and detect possible mutations is needed [10,21].

## 4. Conclusions

Considering the clinical signs, the necropsy findings, the pulmonary changes with a nodular appearance and infiltration of PHCs and the immunohistochemistry results, we confirm the first-ever reported case of a rare histiocytic proliferative disorder in an African lion.

## Figures and Tables

**Figure 1 animals-14-01011-f001:**
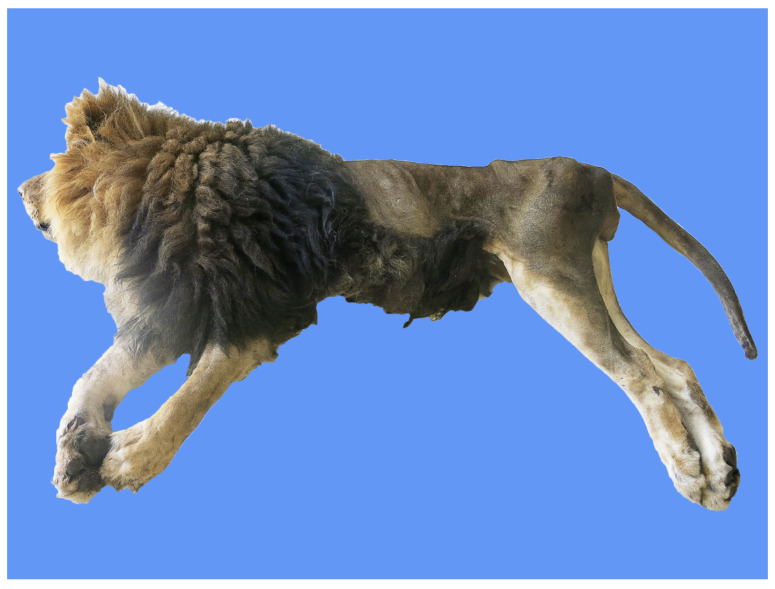
The corpse of the African lion was severely emaciated.

**Figure 2 animals-14-01011-f002:**
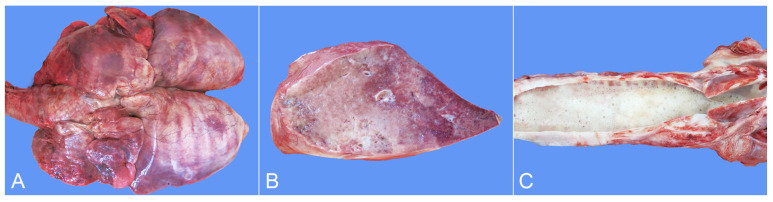
Gross lesions of the lung. (**A**) All the diaphragmatic and accessory lobes and most of the apical and cardiac lobes of the lungs were enlarged and consolidated. The consolidated areas were grayish-white or grayish-red, with a glossy, fleshy appearance. (**B**) The sections of consolidation areas in the lung were grayish-white or grayish-red and rough, with visible different-sized yellowish-white lesions. (**C**) The trachea was filled with a large amount of white foamy fluid.

**Figure 3 animals-14-01011-f003:**
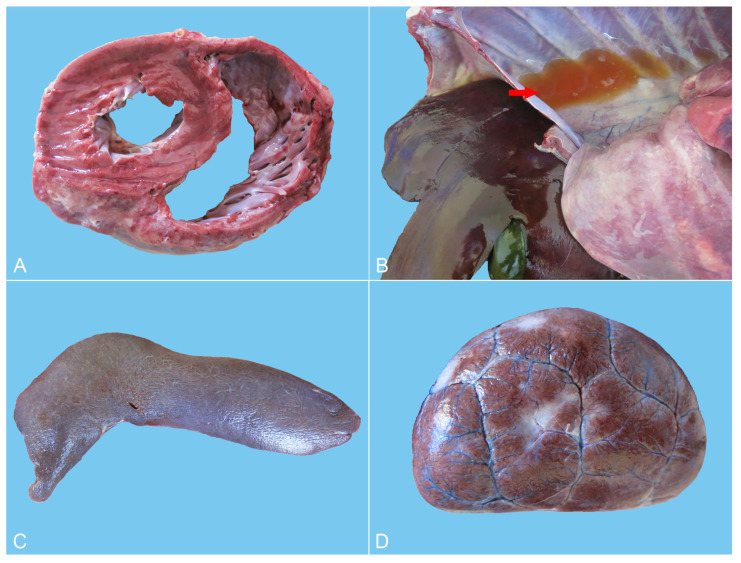
Gross lesions of the heart, liver, spleen and kidneys. (**A**) Severe eccentric hypertrophy with a thickened ventricular wall and significant ventricular dilation. (**B**) The liver was dark red with a smooth surface. In addition, a small amount of light yellow semi-transparent pleural fluid was present in the chest cavity (arrow). (**C**) The spleen was soft with a dark purplish red surface. (**D**) The renal volumes decreased with scattered concave grayish-white lesions of varying sizes on the surface.

**Figure 4 animals-14-01011-f004:**
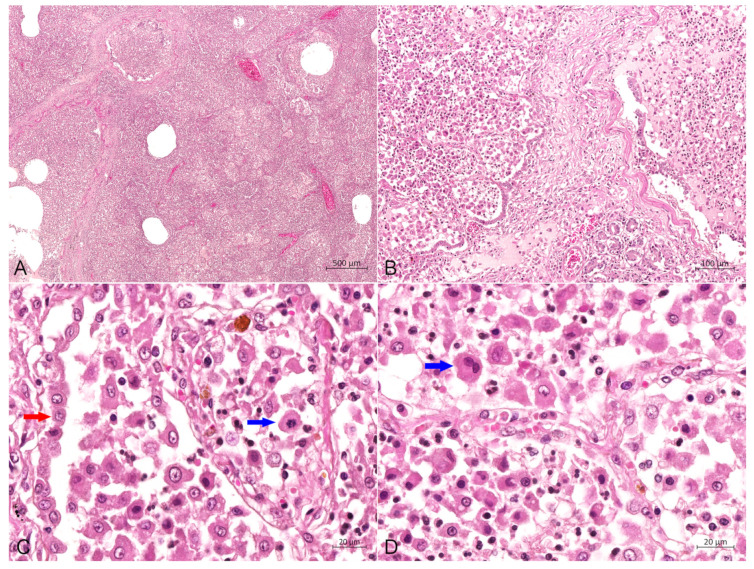
Hematoxylin-eosin stain of the lung. (**A**) The lobular, bronchiolar and alveolar structures of the lung disappeared and were filled with a large number of PHCs, bar = 500 μm. (**B**) Proliferation of numerous histiocytic cells in the alveolar cavity (left) and bronchioles (right), with severe fibrosis in the interstitium, bar = 100 μm. (**C**) The alveolar cavity was repleted of PHCs, with occasional mitotic figures (blue arrow), and type II alveolar epithelial cells (red arrow) showed prominent hyperplasia, bar = 20 μm. (**D**) A large number of PHCs in the alveolar cavity with occasional binucleated cells (blue arrow), bar = 20 μm.

**Figure 5 animals-14-01011-f005:**
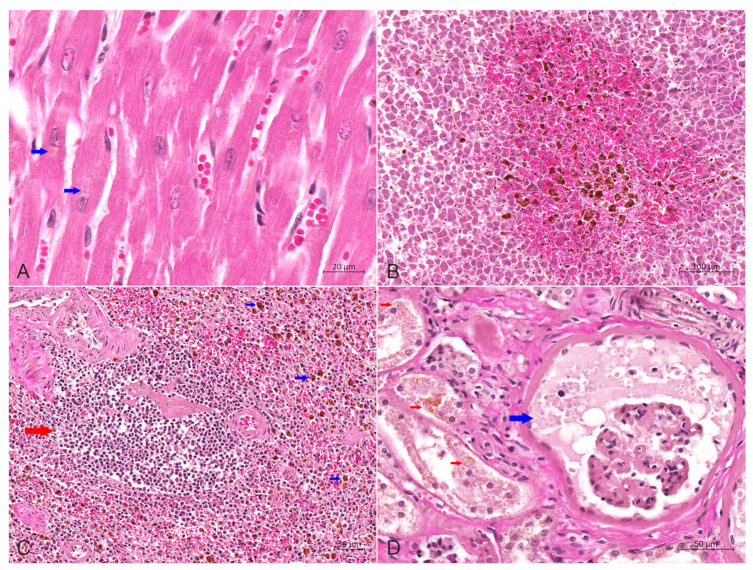
Hematoxylin-eosin stain of heart, liver, spleen and kidneys. (**A**) Slight congestion and dilation of the small blood vessels in the myocardial interstitium were observed, with a small amount of brownish yellow lipofuscin particles (red arrow) in the cytoplasm of myocardial cells, bar = 20 μm. (**B**) The central area of the liver lobules was significantly congested, and the activation and proliferation of Kupffer cells containing hemosiderin were increased, bar = 100 μm. (**C**) The number of lymphocytes in the spleen was reduced considerably, and the volume of white pulp (red arrow) was decreased. Also, macrophages containing hemosiderin (blue arrow) were dispersed in the red pulp, bar = 50 μm. (**D**) Wide renal fibrosis was seen, with the extensive proliferation of connective tissue around some glomeruli, dilation of renal sacs filled with protein-rich urine (blue arrow), degeneration of renal tubular epithelial cells, and noticeable lipofuscin particles (red arrow) in the cytoplasm, bar = 50 μm.

**Figure 6 animals-14-01011-f006:**
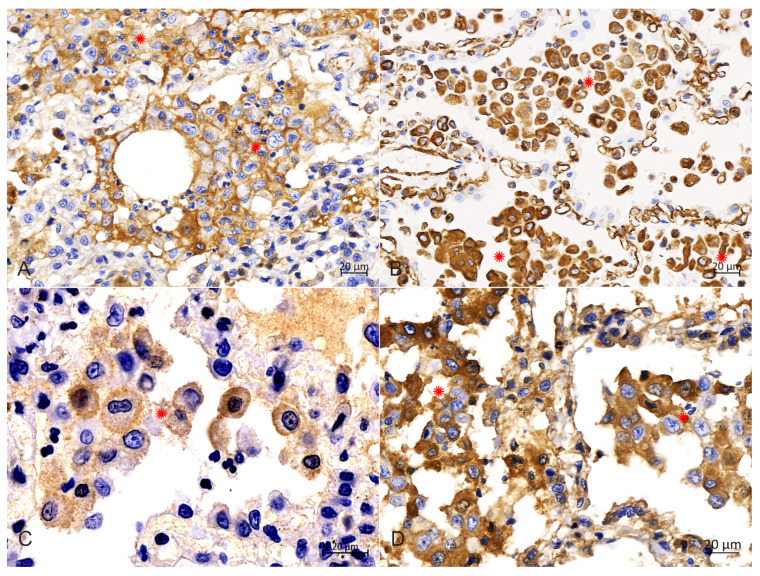
Immunohistochemical analysis in the cytoplasm of PHCs. (**A**) Immunohistochemical staining of the CD1a was strongly positive in the membrane of PHCs (around asterisks), bar = 20 μm. (**B**) Immunohistochemical staining of vimentin was strongly positive in the cytoplasm of PHCs (around asterisks), bar = 20 μm. (**C**) Immunohistochemical staining of S100 was strongly positive in the intramembrane of PHCs (around asterisks), bar = 20 μm. (**D**) Immunohistochemical staining of E-cadherin was strongly positive in the cytoplasm of PHCs (around asterisks), bar = 20 μm.

## Data Availability

All the data supporting the findings of this study are available within the article. All mentioned data are also available from the corresponding author upon request.

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
