# Peer review of "Pulmonary Langerhans Cell Histiocytosis in an African Lion: A Rare Case Report"

_animals, 2024, doi:10.3390/ani14071011_

Round 1
Reviewer 1 Report
Comments and Suggestions for Authors
I've attached the reviewed text. All my suggestions were incorporated into the file.

I have made numerous English grammar and spelling changes without altering the meaning. The changes were incorporated into the attached file.
Reviewer 2 Report
Comments and Suggestions for Authors
The revised manuscript presents a single clinical case of a rare disease in an African lion kept in captivity.
The work is well written and argued and contains information that may be of interest to the scientific-medical community.
This reviewer's main criticism concerns the weakness of the clinical data presented. No consistent information is provided on the duration of the symptoms, their progression, the animal's environment (coexistence with other specimens, relationship). There is no reference to physical examination data. During the time the illness lasted, it is likely that some sedation of the animal was planned to collect exploration data or additional examinations. This is not indicated. In any case, a remote examination should at least reflect data on respiratory rate, respiratory pattern, mucous membrane color, presence of cough, gastrointestinal signs, water consumption and volume of urine production, etc. Nor is the reasoning followed for the selection of the empirical therapies administered in relation to the clinical status and diagnostic suspicion. There is no list of differential diagnoses considered during the process. All these aspects should be reviewed and expanded.
Any reader not familiar with these protocols may wonder how intramuscular injections were performed daily as a dangerous animal. This must also be indicated.
On the contrary, the description and argumentation regarding the anatomopathological, histological and immunohistochemical findings are precise and appropriate, being the main contribution of the manuscript.
Some specific considerations would be:
Line 136: in addition to the descriptive, this section should end with a final conclusive histopathological diagnosis.
Lines 155, 178 and 195; figures 4, 5 and 6: mark the referred structures, key findings and cell types with asterisks, arrows or letters
Line 193. At the end of the presentation of findings, a compilation/conclusive section must be included that argues the specific diagnosis of the disease, explanation for the clinical course and final cause of death.
Line 203. The diagnosis is not based on clinical findings because these are very poor. It is a postmortem diagnosis.
Line 220-223: the choice of this treatment should be discussed and argued, as well as better options if more data were available about what was happening to the animal (for example, pleural drainage, diuretics, bronchodilators).
Line 240-251: the authors refer without explicitly doing so to the pathophysiology of right heart failure secondary to pulmonary hypertension (cor pulmonale) secondary to chronic lung disease. This argument should be enriched along these lines and better support the cardiovascular consequences of lung disease in this individual clinical case.
Reviewer 3 Report
Comments and Suggestions for Authors
In this case report the authors describe the clinical, anatomopathological, histological and immunohistochemical evaluations of Pulmonary Langerhans Cell histiocytosis in an African Lion.
Considering the scarce data in the literature on PLCH in this species, it can certainly contribute to information regarding these aspects.
In general, the manuscript it is well structured, and the exposition is clear, with beautiful photos representative of the clinical case.
However, I would like to suggest some minor revisions to be made, although Minor editing of English language is required.
Manuscript remarks:
Line 25: showing instead of showed.
Line 26: and died of exhaustion, instead of who died.
Line26: illness, it coud be better instead of disease.
Line 27: please erase obviously.
Line 29: replace Furthermore instead of in addition.
Line 29: had scattered greyish-white….
Line 30: while the congestion was widely distributed….
Line 32: bronchioles (please erase the)
Line 33: evident instead of obvious
Line 36: intestine and spleen
Line 9l: the number (21) points out what? Please specify it.
Line 101: replace the hair was depilated with: and extensive alopecic areas were present.
Line 128: moreover, other tissues and organs showed congestion. Please erase obvious and as well
Line 155: Ematoxylin-eosin stain of the lung, instead of Histopathological features. Please insert (Bar=….) after every histological description
Line 167: insert and the volume of….; erase and the number was few
Line 169: they were characterized…
Line 175: other findings included edema and interstitial connective tissue hyperplasia of the adrenal gland.
Line 178: please, insert hematoxylin-eosin stain and the scale bar in the legend (Bar=….µm)
Line 183:……was significantly reduced as the volume of the white pulp.
Line 211: in the case presented here or in this case, the patient…..
Line 212: mental depression
Line 217: with the treatment
Line 218: no clinical response was seen and ultimately all cats died.
Line 251: And finally, the patient died of ipoxia….
Line 257: in this case
Line 258: bronchioles (please erase the)
Line 260: with micotic and binucleated appearance occasionally observed. In addition, the pulmonary septa were markedly widened probably due to the infiltration……
Line 262: please erase the phrase: during the pathological process of this disease and write something like “in our case report” …
Line 263: lung parenchyma
Line 279: in this case
Line 279: were mainly highlighted as
Line 280: please erase obviously.; it is better: evident hyperplasia of the glomerular capsule, probably caused by infiltration……
Line 283: please erase and infiltrated; rewrite: edema and interstitial .c .t .h. were observed in the adrenal gland.
Please rewrite this sentence: The wall and villi of the small intestine had thinned, and the white pulp of the spleen had shrunk significantly.
Line 305: in our study, the positive…..
Line 311: could be ruled out to be monocyte or macrophage.
Line 356 figure legend: again?
Line 394: please rewrite the references with the specific guidelines of the journal.
Comments on the Quality of English Language
Minor editing of English language is required.
Reviewer 4 Report
Comments and Suggestions for Authors
the work is interesting but the diagnosis may not be correct. it needs to be reformulated. The markers don't support the diagnosis. Cd1a positivity is said to indicate Langerhans cells, which is a lie. It can be positive in dermal dendritic cells. SAe has not been reformulated and the conclusions can be hypothetical or if other markers are not used (e.g. cadherin E), it should not be accepted, in my opinion.
Best regards
Author Response
Reviewer #4:
the work is interesting but the diagnosis may not be correct. it needs to be reformulated. The markers don't support the diagnosis. Cd1a positivity is said to indicate Langerhans cells, which is a lie. It can be positive in dermal dendritic cells. SAe has not been reformulated and the conclusions can be hypothetical or if other markers are not used (e.g. cadherin E), it should not be accepted, in my opinion.
We would like to thank the referee for reviewing our work. We accept your comment on the marker detection of Langerhans cells and have added the IHC results of E-cadherin for being more rigorous and precise. The corresponding changes are indicated in red color in the revised manuscript.
Round 2
Reviewer 4 Report
Comments and Suggestions for Authors
I would like to thank the authors for reviewing the article and for including Cadeina E.
Congratulations